# Is this the solution to wellbeing and burnout management for the critical care workforce? A parallel, interventional, feasibility and realist informed pilot randomized control trial protocol

**Nurul Bahirah Binte Adnan**[1]*, **Hila Ariela Dafny**[1], **Claire Baldwin**[1], **Gavin Beccaria**[2,3], **Diane Chamberlain**[1]

**1** Caring Futures Institute and College of Nursing and Health Sciences, Flinders University, Bedford Park, South Australia, Australia, **2** School of Psychology and Wellbeing, University of Southern Queensland, Darling Heights, Queensland, Australia, **3** Institute of Resilient Regions, University of Southern Queensland, Darling Heights, Queensland, Australia

☉ These authors contributed equally to this work.
* Nurul.Adnan@flinders.edu.au

**Data Availability Statement:** No datasets were generated or analysed during the current study. All

## Abstract

Critical care healthcare professionals are at high risk in developing burnout and mental health disorders including depression, anxiety, and post-traumatic stress disorder. High demands and the lack of resources lead to decreased job performance and organizational commitment, low work engagement, and increases emotional exhaustion and feelings of loneliness. Peer support and problem-solving approaches demonstrate promising evidence as it targets workplace loneliness, emotional exhaustion, promotes work engagement, and supports adaptive coping behaviors. Tailoring of interventions have also shown to be effective in influencing attitudes and behavior changes, attending to the individual experience and specific needs of end-users. The purpose of this study is to assess the feasibility and user-perceived acceptability of a combined intervention (Individualized Management Plan (IMP) and Professional Problem-Solving Peer (PPSP) debrief) in critical care healthcare professionals. This protocol was registered in the Australian and New Zealand Clinical Trials Registry (ACTRN12622000749707p). A two-arm randomized controlled trial, with pre-post-follow-up repeated measures intergroup design with 1:1 allocation ratio to either 1) treatment group–IMP and PPSP debrief, or 2) active control group–informal peer debrief. The primary outcomes will be conducted by assessing the recruitment process enrolment, intervention delivery, data collection, completion of assessment measures, user engagement and satisfaction. The secondary outcomes will explore preliminary effectiveness of the intervention using self-reported questionnaire instruments from baseline to 3-months. This study will provide the interventions' feasibility and acceptability data for critical care healthcare professionals and will be used to inform a future, large-scale trial testing efficacy.

relevant data from this study will be made available upon study completion.

**Funding:** The author(s) received no specific funding for this work.

**Competing interests:** The authors have declared that no competing interests exist.

## Introduction

Wellbeing is a combined definition of feeling good and functioning well [1]. It can be conceptualized as a spectrum with high wellbeing, flourishing, and happiness at one end, and low wellbeing, increased anxiety, and depression at the other end [2]. High wellbeing portrays the experience of positive emotions and having a sense of purpose and control [1]. It is a sustainable condition that enables individuals to develop within their environment and thrive [3]. Individuals with high wellbeing in the workplace demonstrate optimal workplace engagement, productivity, and performance [3, 4]. Work engagement is characterized by dedication, vigor, and absorption [5, 6]. It is an indicator of intrinsic motivation and general autonomy within the workplace [6]. Unlike intrinsic motivation that focuses on one job task, work engagement reflects on a more pervasive and persistent affective-cognitive state [6]. Engaged workers will feel enthusiastic, energized, and will want to genuinely work [5, 6]. These workers facilitate positive organizational outcomes as they transcend employment expectations [6].

Burnout and work engagement are negatively associated and influence mental and physical health [5, 7]. Burnout is a prolonged response caused by chronic exposure to interpersonal stressors within the workplace [8]. Maslach and Leiter (2016) defined burnout as a psychological syndrome consisting of three dimensions: exhaustion, feelings of cynicism, and having a sense of ineffectiveness [8]. Exhaustion was described as the loss of energy, wearing out, depletion, fatigue, and debilitation, whereas cynicism portrays irritability, withdrawal, inappropriate attitudes towards patients, and loss of idealism [8]. Inefficacy was depicted as low morale, reduced capability or productivity, and the inability to cope [8]. Burnout and work engagement are influenced by job characteristics, which can be conceptualized by the job demands and job resources model (Job Demands-Resources (JD-R) model) [7, 9]. Job demands are energy-depleting aspects of the job that requires emotional, physical, and cognitive efforts—for example, workload, role stress, and work pressure [5, 9]. When employees are exposed to high job demands, they become chronically exhausted and psychologically distances themselves from their work [9]. Available and adequate job resources may psychologically fulfil the needs of employees and buffer the effects of high job demands on burnout [5, 9]. Job resources aims to achieve work goals and encourage personal development and growth [5, 9]. The JD-R model also considers the use of personal resources to attend to job demands [9]. Personal resources refers to self-beliefs, for example, the development of self-efficacy, optimism, and resilience—as it motivates employees to achieve work-related goals, which improves their wellbeing and job performance [9].

Healthcare professionals encounter high workload, adversities, and stress on a daily basis, which makes them vulnerable to burnout [10]. While burnout is experienced by healthcare professionals in general, the critical care workforce has been identified to be at heightened risk [11]. This is because critical care healthcare professionals are exposed to difficult daily workloads, high patient trauma, mortality, tragedy, and encounter challenging ethical situations on a daily basis [12]. The coronavirus 2019 (COVID-19) global pandemic has added a greater magnitude of strain on the critical care workforce with higher patient acuity, mortality rates, and struggles for personal safety [13]. National leaders have raised concerns regarding the impact of COVID-19 on the workforce's wellbeing [13]. A systematic review of critical care healthcare professionals revealed 49.3% to 58% of the workforce experienced burnout and many suffer from mental health disorders [14]. Another cross-sectional study demonstrated 41% of intensive care workers had low wellbeing and 46% had peritraumatic distress [11]. Psychiatric morbidities including depression, anxiety, insomnia, post-traumatic stress disorder, suicidal ideation, and somatization were also significant [15–17].

Reports suggests elevated levels of loneliness during the pandemic for healthcare professionals [18]. Loneliness is prominent amongst healthcare professionals, where a combined prevalence across 101 countries was 21% during the COVID-19 pandemic in comparison to 6% before the pandemic [18]. It is a major health concern as it is a predictor of high emotional exhaustion of workplace burnout [19, 20]. Loneliness occurs when employees begin to lack meaningful social relationships regardless of the number of contacts [20]. It focuses on the quality of relationships and inconsistent expectations between desired and real social relations [20]. Loneliness and social isolation have drastic implications in the workplace, contributing to burnout and decreased job performance, creativity, organizational commitment, and work engagement [21, 22]. Work engagement is demonstrated as mediator between loneliness and organizational citizenship behaviors, which are relative to job satisfaction and employee wellbeing [21–23].

Critical care employees working in extremely difficult and high-risk conditions require urgent need for mental support [20]. Peer support is an important element and a significant work resource with positive effects on mental health [20]. It refers to the sharing of common experiences by individuals facing similar challenges and giving and receiving help based on knowledge from the shared experience [24]. Peer support can facilitate the development of decision-making, problem-solving, coping and stress management skills [25]. Supports from co-workers and friends have demonstrated reduction in emotional exhaustion, burnout experiences and loneliness [20, 26]. Maslach and Goldberg (1998) suggests that peer support groups may provide new insights, personal rewards, emotional comfort, and may contribute as a source for optimism, humor, and encouragement during difficult and stressful times [26, 27].

Problem-solving abilities significantly influences overall psychological wellbeing and social competence because the ability to resolve or cope with daily stressors is strongly affiliated to social and personal functioning [28, 29]. Problem-solving can be viewed as a self-directed cognitive-behavioral process, where individuals attempt to discover or identify adaptive or effective solutions to a problem encountered during their daily living [29]. Ineffective or maladaptive coping behavior leads to various personal and social consequences (i.e., anxiety, anger, psychological distress) [29]. Contrastingly, problem-solving skills can act as a buffering factor to attenuate the negative effects of stress [29].

Tailoring is a process of customizing to match the characteristics of individuals [30]. It is used to respect the differences amongst people, which influences attitudes and facilitates behavior change by attending to the specific needs of end-users [30, 31]. By understanding tailoring, and the aforementioned research and practice context for critical care healthcare professionals, our research team were prompted to conduct an umbrella review that sought individual interventions to improve wellbeing and decrease burnout for critical care healthcare professionals [32, 33]. This was followed by realist expert opinion paper by Adnan et al. (2022) that interviewed 21 critical care experts to describe (using theory prepositions) contextual factors and mechanisms required for an intervention to work for critical care healthcare professionals [32]. Tailoring was highlighted as an essential component in both the umbrella review and expert opinion methodologies, which enabled development and refinement of a program theory of individual-focused interventions for healthcare professionals in critical care settings [32, 33]. The authors then used the program theory to guide the design of the intervention in this study.

This study will combine the three concepts of peer support, problem-solving and tailoring to determine its effectiveness in improving wellbeing and engagement and decreasing burnout amongst critical care healthcare professionals. To the authors' knowledge, there are no similar studies with the proposed intervention within the critical care workforce. Thus, the primary aim of this study is to test the feasibility and user-perceived acceptability of a combined

Individual Management Plan (IMP) and Professional Problem-Solving Peer (PPSP) debrief in comparison to informal peer debrief. Specifically, the study will test the feasibility of the recruitment process, enrolment, intervention delivery, data collection, and completion of assessment measures in a sample of critical care healthcare professionals. User-perceived acceptability will be assessed using participant engagement to the intervention and user satisfaction. The secondary aim is to examine the preliminary effectiveness of the intervention on improving wellbeing, work engagement, and decreasing burnout. This study hypothesizes that the combined intervention will be (1) feasible (in terms of recruitment, enrolment, intervention delivery, and data collection, and completion of assessment measures), (2) acceptable in terms of participant engagement and user satisfaction, and (3) effective in improving wellbeing, engagement, and decreasing burnout symptoms than the informal peer debrief intervention at post-treatment and at the two follow-up periods (1 month and 3 months).

## Materials and methods

### Study design

This study will be a parallel-designed, two-armed, pilot randomized control trial (RCT) determining and comparing the feasibility and user-perceived acceptability of a personalized management program (IMP and PPSP) and conventional program (informal peer debrief) in critical care healthcare professionals. It will also determine the preliminary effectiveness of improving wellbeing, work engagement, and decreasing burnout symptoms. After baseline screening questionnaires are completed, eligible participants will be randomized into one of two groups in a 1:1 ration. This protocol was prepared in accordance with the SPIRIT guidelines (Fig 1 and S1 Table). Flinders University (ref. number: 4703) and University of Southern Queensland (ref. number: H22REA178) Human Research Ethics Committee approved the study (S2 Table). The study was also registered under the Australian and New Zealand Clinical Trials Registry (ACTRN12622000749707p). Fig 2 demonstrates the study flowchart.

### Study setting

The study center will be located at the College of Nursing and Health Sciences, Flinders University. Study data will be collected through online questionnaires using Qualtrics [34] and virtual sessions (IMP, debrief, and feedback semi-structured interviews) using Zoom Video Communications [35].

### Eligibility criteria

Critical care professionals who are interested to participate will register their interest in an electronic form. The form will contain the Impact of Event Scale-Revised (IES-R), which will be used to assess their eligibility prior to completing and signing the electronic informed consent form. The eligibility criteria will include (a) registered healthcare professionals practicing within an Australian or New Zealand critical care setting (i.e., Physicians, Nurses, Allied Health) and (b) have access to a personal computer or device with camera and microphone. The trial will also include individuals that are either undertaking or not undertaking prescribed treatments for mental health conditions. The exclusion criteria will include (a) individuals that are under any type of work compensation claims, (b) have contradictions to any of the interventions, and (c) are considered as 'high risk' of self-harm or psychological harm, which will be determined at the recruitment phase using the Impact of Event Scale-Revised (IES-R) [36]. Potential participants scoring 37 or more on the IES-R will be excluded, as this demonstrates a score that is high enough to suppress the individual's system's functioning,

| | Enrolment | Allocation | During treatment | | | | | | | | | End treatment | | |
|---|---|---|---|---|---|---|---|---|---|---|---|---|---|---|
| **STUDY PERIOD** | | | | | | | | | | | | | | |
| **TIMEPOINTS →** | | **T0** | **T1** | | | | | | | | | **T2** | **T3** | **T4** |
| **Week →** | | | **1** | **2** | **3** | **4** | **5** | **6** | **7** | **8** | | | |
| **ENROLMENT:** | | | | | | | | | | | | | | |
| *Eligibility screen* | X | | | | | | | | | | | | | |
| *Informed consent* | X | | | | | | | | | | | | | |
| *Randomization* | | X | | | | | | | | | | | | |
| *Allocation* | | X | | | | | | | | | | | | |
| **INTERVENTIONS:** | | | | | | | | | | | | | | |
| *IMP + PPSP debrief** | | | ←→ | | ←————————→ | | | | | | | | | |
| *Informal peer debrief* | | | | | ←————————→ | | | | | | | | | |
| **ASSESSMENTS:** | | | | | | | | | | | | | | |
| *Sociodemographic data* | X | | | | | | | | | | | | | |
| *IES-R* | X | | | | | | | | | | | | | |
| ***Primary outcomes*** | | | | | | | | | | | | | | |
| *Journaling* | | | X | X | X | X | X | X | X | X | | | | |
| *Post-treatment feedback (questionnaire & interview)* | | | | | | | | | | | X | | | |
| ***Secondary outcomes (Psychometric scales)*** | | | | | | | | | | | | | | |
| *CD-RISC* | X | | | | | | | | | | X | X | X | |
| *TIS* | X | | | | | | | | | | X | X | X | |
| *MBI* | X | | | | | | | | | | X | X | X | |
| *PES* | X | | | | | | | | | | X | X | X | |
| *DASS-21* | X | | | | | | | | | | X | X | X | |
| *PSI* | X | | | | | | | | | | X | X | X | |
| *PANAS* | X | | | | | | | | | | X | X | X | |

**Fig 1. SPIRIT schedule of enrolment, interventions, and assessments.** *IMP will run for 2 weeks (blue arrow), PPSP debrief will run for the following 6 weeks (black arrow), IMP, Individualized Management Plan; PPSP, Professional Problem-Solving Peer; IES-R, Impact of Event Scale-Revised, CD-RISC, Connor-Davidson Resilience Scale 25; TIS, Attrition Turnover Inventory; MBI, Maslach Burnout Inventory; PES, Practice Environment Scale; DASS-21, Depression, Anxiety, and Stress Scale-21; PSI, Problem Solving Inventory; PANAS, Positive and Negative Affect Schedule.

requiring referral for professional support [36, 37]. Participants that score 37 and above will be contacted by the research team and offered a one-on-one consult with a clinical psychologist, where they will be on-referred to an employee assistance program or their general practice for further professional support.

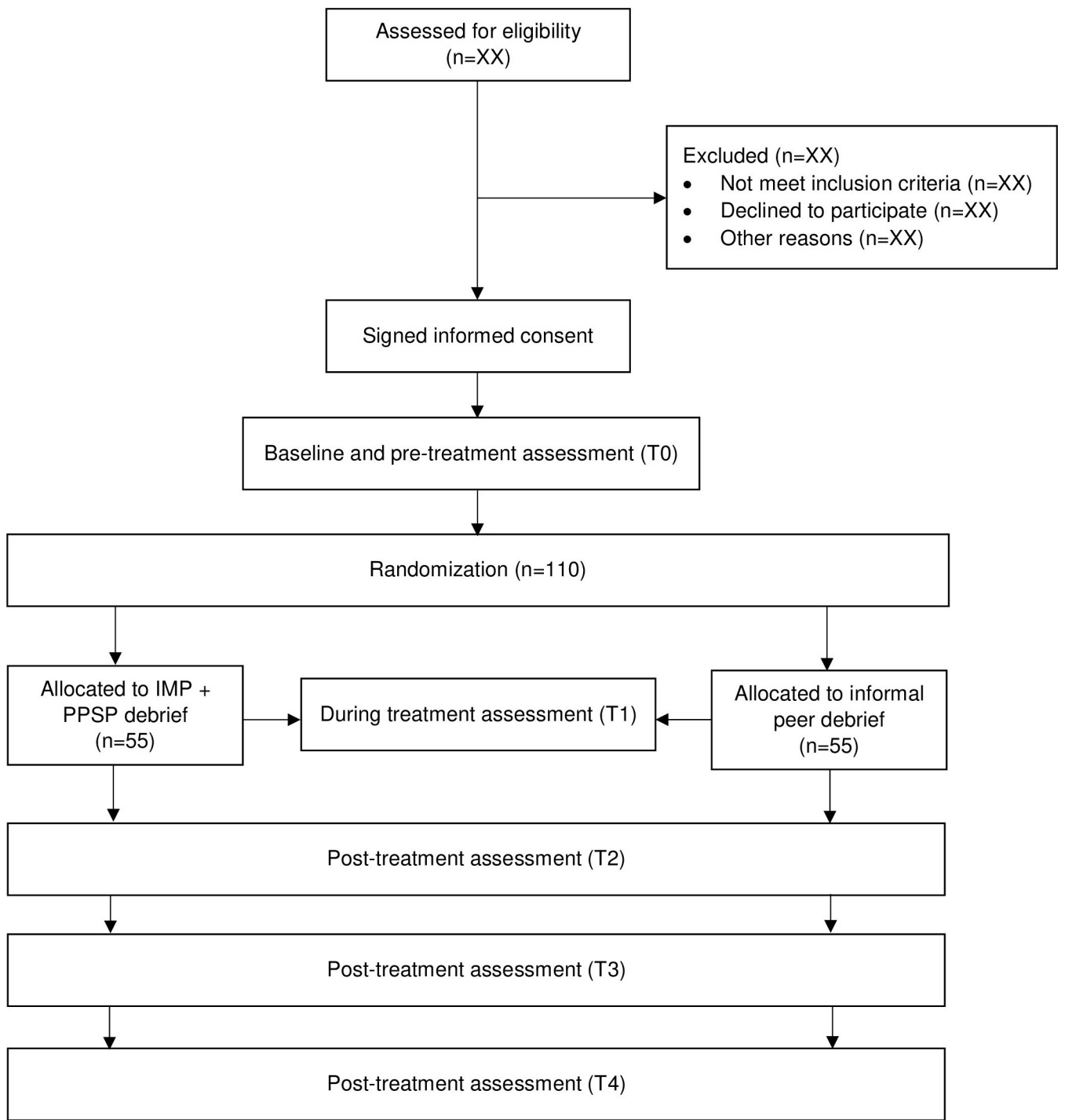

**Fig 2. Study flowchart.**

## Intervention

**Intervention group.** The intervention group will receive an Individualized Management Plan (IMP) and Professional Problem-Solving Peer (PPSP) debrief. The IMP is a 30-minute,

virtual, once-off, one-to-one consult with a psychologist. The psychologist will provide tailored, immediate, and simple strategies within the session as their usual scope of practice (using clinical judgement). These strategies will be centered around the participant's everyday living, for example, recommendations for improved sleeping habits, encourage development of exercise habits, and methods for relaxation. Participants will be advised by the psychologist to implement these strategies into their daily lives. The list of semi-structured questions can be found in Table 1.

This study will administer a professional peer debrief led by accredited peer support leader (s). It will include a semi-structured, 6-week, group-based, virtual, informal, peer debrief session. The peer debrief session aim to encourage social interactions and engagement between participants rather than to achieve structured debrief objectives, such as successful reflection. The objectives are weighted towards achieving peer support using informal structures of debrief. The peer debrief will also incorporate the theory of social problem-solving. The goal for peer debrief is to learn and apply problem-solving approaches [38]. The peer support leader will support participants to positively change problematic situations and/or provide techniques to decrease emotional distress from the afflicted problem [38]. This will be facilitated through group discussions, experience sharing, and individual guidance to summarize the causes of the problem, develop feasible goals and strategies, and perform analysis on the effects (success or failure) of the implemented strategy [38].

The purpose of integrating IMP was due to the key component of 'tailoring', which was identified in Adnan et al. (2022) refined program theory [32, 33]. Tailoring caters towards the individual's experience, addressing contextual factors and mechanisms. It also facilitates understanding of the unique characteristics and experiences of the individual, which may allow investigators to address their individual's needs, resources, preferences, and goals; ultimately giving meaning to the intervention, increasing personal relevance, and prioritizing the user's control and involvement. Moreover, debrief facilitates the reflection of situations, exploring one's thoughts, and questioning assumptions to achieve positive learning that is based on lived experiences [39]. An individual's performance can also be revisited to enable greater understanding of the situation, foster critical thinking, and provide adjustments toward future situations [39].

**Table 1. IMP semi-structured questions.**

| Themes | Guiding Questions |
| --- | --- |
| General Questions | How are you feeling? |
| | How are you finding your job at the moment? Are there any stressors that you are currently experiencing? |
| Mental Health | Can you tell me about any times over the past few months/weeks that you've been bothered by low feelings, stress, or sadness? |
| Physical Health | Has it affected your sleeping habits? Have you noticed any changed? Difficulty sleeping? Restlessness? |
| | How would you describe your appetite over the past weeks? Have your eating habits changed in any way? |
| Autonomy, choice, and control | How often during the past months/weeks have you felt as though your moods, or your life, were under control? |
| Relationships and belonging | Describe how 'supported' you feel by others around you—friends, family or otherwise |
| Self-perception | Let's talk about how often you have felt satisfied with yourself over the past months/weeks. |
| Hope and hopelessness | Can you tell me about your hopes and dreams for the future? What feelings have you had recently about working towards those goals? |

**Active control group.**    The informal peer debrief will include mutual support between participants with shared experiences. Like PPSP debrief, its goals are not to achieve debrief outcomes, rather, to encourage social interaction and engagement between participants— enabling a platform to discuss concerns, problems, and feelings [40]. The informal peer debrief will be a semi-structured, 6-week, group-based virtual, peer debrief intervention. Peer support leader(s) will conduct the sessions but will not provide active solutions. The list of semi-structured questions will be consistent in both intervention and control debrief groups with differences in the 'Pre-briefing', 'Discovering', and 'Deepening' sections; informal peer debrief will not include problem-solving approach (see Table 2).

## Measurements

Participants will complete self-reported questionnaire measures at four timepoints: baseline (T0), after treatment (T2), 1-month follow-up (T3), and 3-months follow-up (T4). Qualitative journal entry will be completed throughout treatment (T1)—the journal entry will include reflective questions for participants complete. Participants will additionally provide sociodemographic characteristics at T0 and treatment feedback (both through an online questionnaire and demi-structured interviews) at T2. Examples of sociodemographic characteristics will include gender, age, living status, citizenship, employment, qualifications, and family responsibilities. Audits of study screening, enrolment logs, attendance list to the consult and debrief sessions (intervention and active control), and study records (completion of the assessment measures at the five assessment points) will also be conducted.

## Primary outcomes

**Feasibility.**    Feasibility of the study will be assessed based on the recruitment process, enrolment, treatment delivery, and data collection. The audits of participants referred and attended the consults each week will be used to measure the recruitment and enrolment. Audits of completing questionnaires, journal entries, feedback questionnaires, and feedback

**Table 2. Debrief session layout.**

| Topic | Content |
|---|---|
| Introduction | • Introduction of the support leader and researcher present in the session<br>• What this study is about and who is conducting the study |
| **Pre-briefing** | • Purpose of the debrief session and the role of the support leader in facilitating discussions and prompting self-reflection<br>• Items that can be discussed (i.e., encounters/stressors at the workplace)<br>• **Management of situations using problem-solving approaches and connecting new learnings into future situations.** |
| Defusing | • Encouraging participants to talk about how they feel when encountered with a stressful situation.<br>• Allow reflection of their feelings and recapping the scenario |
| **Discovering** | • Allow other participants to reflect on the situation and provide their thoughts and ideas<br>• Enable participants to explore any similar situations that they may have encountered.<br>• **Support leader(s) may ask other participants on how they would approach the situation, their rationale, and potential future strategies. The support leader(s) can also offer their own mental model on problem-solving approaches to deal with the situation.** |
| **Deepening** | • **Reflection on what can be done in similar future situations.**<br>• **Finding a relationship between the suggested strategy to other situations at work.** |
| Summary | • Recapping what was discussed in the session<br>• Ending with one thing that can be taken away from the session and used in their workplace. |

Note: The intervention group (PPSP debrief) will also use techniques stated in both 'roman' and 'bold' typeface, whereas the control group will follow techniques noted in 'roman' typeface only.

semi-structured interviews will measure the feasibility of treatment delivery and data collection.

**User-perceived acceptability.** User-perceived acceptability will measure user intervention engagement and user satisfaction. Both outcomes will be measured using journal entries, treatment feedback questionnaire, and feedback semi-structured interviews. Journal entry question such as 'What knowledge/strategies were you able/unable to implement from the debrief session?" will measure intervention user engagement. Treatment feedback questionnaire will be used to measure user satisfaction, which includes questions surrounding user experience, barriers, enablers, and likelihood of recommendation. This will be further supported by feedback semi-structured interviews, that will be conducted as a 30 minute, one-on-one, once-off session for each participant. The feedback semi-structured interviews aim to ask questions surrounding feasibility and user-perceived acceptability. The questions will include topics that cover benefits, improvements, and potential changes for the future larger study. Other secondary outcomes will be measured using questionnaire instruments described below.

## Secondary outcomes

The secondary measures are directed to determine the impact of the intervention. Secondary outcome measured using self-reported questionnaire instruments, treatment feedback questionnaire, feedback semi-structured interviews, and journal entry. The journal entry question on 'I feel burned out' will measure the participant's perception on burnout. The treatment feedback questionnaire and feedback semi-structured interviews that addresses questions surrounding benefits of the intervention and changes in participant's viewpoints measures the impact of the intervention.

**The Utrecht Work Engagement Scale (UWES).** The UWES-9 is used to measure work engagement–assumed to be the opposite of burnout. It is a 9-item revised scale scored on a 7-point Likert scale from "Never" to "Always" [41].

**Attrition Turnover Inventory (TIS).** The TIS scale is used to determine turnover intentions [42]. It is a 6 item scale scored on a five point Likert scale, between poles of intensity from 1 being never to 5 being always [42].

**Maslach Burnout Inventory (MBI).** The MBI scale is the gold standard in measuring burnout [43]. It comprises of three scales of emotional exhaustion (9 items), depersonalization (5 items) and personal accomplishment (8 items). It is a 22-item instrument [43].

**Practice Environment Scale (PES).** The PES is an instrument, which measures the practice environment, defined as factors that enhance the ability to practice skillfully and deliver high quality care [44].

**Depression, Anxiety, and Stress Scale (DASS-21).** DASS-21 is a self-reporting scale that is designed to measure negative emotional states of stress, anxiety, and depression with seven subscales within each measure [45]. Scoring can be presented as total scores and scores for three subscales [45]. The subscales are further divided into five severity ranges including normal, mild, moderate, severe, and extremely severe [45].

**Problem-Solving Inventory (PSI).** The PSI assess awareness and evaluates the individual's problem-solving styles or abilities [46]. It is a self-reported measure consisting of 35 six-point Likert scale with three factors including problem-solving confidence, approach-avoidance style, and personal control [46]. Lower scores demonstrate attitudes and behaviors that are associated with successful problem solving [46].

**Positive and Negative Affect Schedule (PANAS).** The PANAS measures positive and negative affect using 20-items; 10 negative affect markers, and 10 positive affect markers [47].

The items are rated on a scale from one being 'very slightly or not at all', to five being 'extremely' [47].

## Sample size estimation

To achieve the primary aims of determining feasibility and acceptability, a pragmatic approach will be taken in that if sufficient data to address the secondary aim (preliminary effectiveness) can be achieved, this should also be sufficient for the primary aim. Power calculations were therefore conducted to determine the sample size required for the secondary aim of this study.. Analyses variance with repeated measures between and within the design in relation to the participants wellbeing was used, with low effect size estimated ($\eta^2 = 0.061$). Significance level was 0.05 and power at 0.95. This resulted to a minimum sample of 106 participants for the study, having considered 20% attrition rate. The sample size was rounded off to 110 participants (55 each group) to enable equal samples sizes in both groups.

## Recruitment

Recruitment will be conducted using flyers distributed to members of the Australian and New Zealand Critical Care Society (ANZICS) and Australian College of Critical Care Nurses (ACCCN). The target audience for recruitment will include critical care healthcare professionals within the organizational membership, who are then able to forward onto their networks and people who may not be members. The flyer will include a link to the study's website containing further information about the trial and a registration form to be completed by interested participants.

## Informed consent

Those interested in participating will be able to register for the study through the study's website (www.wellbeingandburnout.com) or through the distributed flyers. Additional information about the study (i.e. purpose, significance, study content, risks and benefits, and confidentiality principles) can also be accessed through the study's website. Questions can also be asked through study website's platform, as well as through emails, text messages, and phone calls. Participants that register for the study will be provided with an electronic version of the information sheet and consent form to be signed and returned prior to being accepted into the study.

## Group allocation and blinding

The project's work group are experienced in research project management and will be responsible for the recruitment and intervention delivery in both intervention and active control groups. Randomization will be implemented using a randomization table created by a computer software, conducted by an independent research assistant who will be unaware of the characteristics of the study and will not have any involvement in the trial or access to the trial data. This will be a single-blinded study, where the outcomes of assessment will be blinded, however, participants will be aware of their group assignment. Members of the project's work group that are involved in the assessment of participants will be blinded to the participant's assignment. Additionally, since participants will have direct access to their intervention, participants will not be blinded to their group allocation, but will not know which intervention group is active.

## Data collection methods

Baseline (questionnaire scale, sociodemographic, work-related information), journal entry, post-treatment feedback, post-intervention (questionnaire scales), and one- and three-months follow-up (questionnaire scales) assessments will be performed by sending a survey link to participant's email. Participants will be able to complete the questionnaires on an electronic device, such as a computer, tablet, or smartphone. The questionnaires will be checked weekly after each delivery to ensure participant's completion. Participants that do not complete the questionnaires within a specified time will be sent a reminder via email. Likewise, if participants do not attend their Individual Management Plan and/or debrief sessions, the research team will contact participants via email to join the session or to re-book into another session during the week. The feedback semi-structured interviews will be conducted by inviting participants to attend a virtual one-on-one semi-structured interview, which will be organized through email communication.

## Data management

All data files including consent forms, intervention materials, participant's data, consult and debrief notes, questionnaire data, and journal entry data will be stored in the Flinders University managed on-site Enterprise Storage. After the last data collection timepoint (T4), any identifiable data will be de-identified and stored for five years after publication of the results. Following the required data storage period, all data will be securely destroyed. Deidentified research data will be made publicly when the study is completed and published. Feedback of the study outcomes will be provided to participants via a short summary of the research outcomes and a copy of the published research project.

## Data monitoring

A data monitoring committee is not necessary for this study as the planned intervention is short and we are not expecting any harms or adverse effects during the intervention. An independent Safety Officer (researcher who is independent from the study team) will monitor the participant's safety (from participant recruitment to the end of intervention delivery), scientific integrity, study risks and benefits, and ethical conduct of the study. There are no serious adverse events expected, but there may be potential adverse events that could occur during the Individual Management Plan and debrief sessions such as negative emotional reactions and mild distress. If an adverse event were to occur, the safety officer will be notified immediately.

## Planned analysis

Since this study primarily seeks to determine the feasibility and user-perceived acceptability of the intervention, the analysis will focus on key parameters required to conduct a future larger trial. Majority of the analysis will be reported descriptively. The study will measure feasibility of the recruitment, enrolment process, treatment delivery, and data collection. The study screening log, enrolment log, attendance list and study records will be used to descriptively inform these outcomes. User-perceived acceptability of engaging and satisfaction of the intervention will be measured and reported descriptively using journal entries, treatment feedback questionnaire, and feedback semi-structured interviews. Journal entries, treatment feedback questionnaires, and feedback semi-structured interviews will also be analyzed qualitatively using thematic analysis. Thematic analysis will be conducted first manually utilizing Braun and Clarke's (2019) [48] six-step process and with NVIVO 10 software [49]. Two researchers will validate the analysis, and any disagreements will be resolved using a third researcher.

Using thematic analysis will facilitate an essentialist method that enables reporting of meanings, experiences, and the participant's reality, which fits well with the aims and objectives of this study (determining feasibility and acceptability) [48, 49]. The process of thematic analysis will also enable the reporting of secondary outcome measures on the impact of the intervention (i.e., benefits of the intervention, changes in participant's viewpoints, and feeling of burnout).

Secondary outcome measures using questionnaires will be analyzed following the Intention to Treat (ITT). ANOVA or chi-square will be used to compare the sociodemographic data and baseline outcome measures (T0) [50]. Secondly, longitudinal changes (baseline, post-treatment, and 1 month and 3 months follow-up) and the differences between the two groups will be examined using mixed linear models as these models are more precise than the repeated measures of ANOVA [50]. Missing or incomplete data will be considered in the ITT analysis using maximum likelihood estimation method [50]. Effect sizes will be calculated using Cohen's $d$ (bias corrected) to represent the difference between Standard Mean Changes (MC) (T0-T2, T0-T3, T0-T4) [50]. Calculation of SMC in each group will be conducted, providing the $d$ index of the general effect size from the differences between combined Individualized Management Plan (IMP) and Professional Problem-Solving Peer (PPSP) debrief vs. informal peer debrief [50]. After the data is analyzed, summary tables will be provided for all planned evaluations at pre-treatment (T0), post-treatment (T2), 1 month follow-up (T3), and 3 months follow-up (T4) [50]. Results will be reported as frequencies and descriptive statistics (mean, standard deviation, and percentages), summarizing the characteristics of the total sample and participants within each group [50].

## Discussion

Critical care healthcare professionals experience high levels of burnout and mental health implications such as depression, anxiety, and post-traumatic stress disorder [51]. This is relative to chronic occupational stressors that they routinely experience, including high workloads and frequent encounters with adversities [51]. High demands and insufficient resources within critical care implicate to loneliness, poor work engagement and increased levels of burnout symptoms [5, 21, 22]. Amongst the different individual interventions (personal resources) developed to address the symptoms of burnout, peer support demonstrates promising evidence as it targets workplace loneliness, emotional exhaustion, and promotes work engagement [20, 26]. The integration of a problem-solving approach aims to enhance the peer support component through cognitive-behavioral processes to support adaptive coping behaviors [29]. The intervention in this study is also supported by evidence from an expert realist opinion paper (under review) interviewing 21 critical care experts on effective individual interventions. The paper suggested interventions should promote knowledge and skill development, be evidence-based, accessible, inclusive, collaborative, tailored, and promote engagement–which are reflected in the intervention of this study. Tailoring is reflected in the Individualized Management Plan (IMP), which aims to cater to the individual experience and specific needs of end users. The mechanism of self-regulation proposed in the expert opinion was also employed in this study through the integration of a problem-solving approach. Despite strong evidence and theory synthesis supporting the use of these interventions, there have yet to be randomized trials testing and confirming its feasibility, acceptability, and efficacy.

This protocol provides information on a randomized control trial that plans to carry out the intervention based on the combination of tailoring the intervention, peer support, and problem-solving approach on critical care healthcare professionals. The lack of evidence on

the feasibility and acceptability of such concepts requires a pilot study to gain greater insight into the preparation process. It is expected that the participants assigned to the intervention group will demonstrate feasibility and user-perceived acceptability of the treatment processes, alongside statistically significant better results in improving wellbeing, work engagement, and decreasing burnout components. The outcomes in this study will be used as preliminary data for the larger and more defined randomized control trial.

One strength identified in this protocol is the inclusion of a large sample size, which is a result from the estimation of the sample size calculations. This provides an opportunity to acquire more accurate mean values, provides smaller margin or error, and identify outliers that may be present (in skewing the data) within smaller samples [52]. The use of randomization, controlled, and the use of two follow-up timepoints in the design of this study may also provide understanding of both immediate and early longitudinal patterns and outcomes of treatment changes of the intervention. A limitation of the study includes the use of an active control intervention, rather than the use of a waitlist control design. Active controls have ethical advantages as it allows the provision of care to participants that are seeking help and enables non-intervention evaluation. However, such designs may produce overestimates of the intervention's effects and that having an active control counteract and minimize such aspects. Despite this, a greater understanding of the feasibility and acceptability of combining IMP and PPSP debrief is essential to foster a larger study that determines its effectiveness in improving wellbeing, work engagement, and decrease burnout symptoms for critical care healthcare professionals.

## Supporting information

**S1 Table. SPIRIT checklist.**
(DOC)

**S2 Table. Study protocol.**
(PDF)

## Author Contributions

**Conceptualization:** Nurul Bahirah Binte Adnan, Hila Ariela Dafny, Claire Baldwin, Gavin Beccaria, Diane Chamberlain.

**Formal analysis:** Nurul Bahirah Binte Adnan.

**Investigation:** Nurul Bahirah Binte Adnan, Gavin Beccaria.

**Methodology:** Nurul Bahirah Binte Adnan, Hila Ariela Dafny, Claire Baldwin, Diane Chamberlain.

**Project administration:** Nurul Bahirah Binte Adnan.

**Resources:** Nurul Bahirah Binte Adnan, Diane Chamberlain.

**Software:** Nurul Bahirah Binte Adnan.

**Supervision:** Nurul Bahirah Binte Adnan, Hila Ariela Dafny, Claire Baldwin, Diane Chamberlain.

**Validation:** Nurul Bahirah Binte Adnan, Hila Ariela Dafny, Claire Baldwin, Gavin Beccaria, Diane Chamberlain.

**Visualization:** Nurul Bahirah Binte Adnan, Hila Ariela Dafny, Claire Baldwin, Gavin Beccaria, Diane Chamberlain.

**Writing – original draft:** Nurul Bahirah Binte Adnan.

**Writing – review & editing:** Nurul Bahirah Binte Adnan, Hila Ariela Dafny, Claire Baldwin, Gavin Beccaria, Diane Chamberlain.

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
