## [Decision Letter · Decision Letter 0]

9 Mar 2023

PONE-D-22-32113Is this the solution to wellbeing and burnout management for the critical care workforce? A parallel, interventional, feasibility and realist informed pilot Randomized Control Trial protocolPLOS ONE

Dear Dr. Adnan,

Thank you for submitting your manuscript to PLOS ONE. After careful consideration, we feel that it has merit but does not fully meet PLOS ONE’s publication criteria as it currently stands. Therefore, we invite you to submit a revised version of the manuscript that addresses the points raised during the review process.

We look forward to receiving your revised manuscript.

Kind regards,

Ali B. Mahmoud, Ph.D.

Academic Editor

PLOS ONE

Journal Requirements:

2. We note that the original protocol that you have uploaded as a Supporting Information file contains an institutional logo. As this logo is likely copyrighted, we ask that you please remove it from this file and upload an updated version upon resubmission.

Reviewers' comments:

Reviewer's Responses to Questions

**Comments to the Author**

1. Does the manuscript provide a valid rationale for the proposed study, with clearly identified and justified research questions?

Reviewer #1: Yes

Reviewer #2: Yes

2. Is the protocol technically sound and planned in a manner that will lead to a meaningful outcome and allow testing the stated hypotheses?

Reviewer #1: Yes

Reviewer #2: Yes

3. Is the methodology feasible and described in sufficient detail to allow the work to be replicable?

Reviewer #1: Yes

Reviewer #2: Yes

4. Have the authors described where all data underlying the findings will be made available when the study is complete?

Reviewer #1: Yes

Reviewer #2: Yes

5. Is the manuscript presented in an intelligible fashion and written in standard English?

Reviewer #1: Yes

Reviewer #2: Yes

6. Review Comments to the Author

You may also provide optional suggestions and comments to authors that they might find helpful in planning their study.

Reviewer #1: The manuscript describes the protocol of a two-armed parallel, interventional, feasibility pilot RCT assessing the efficacy of an intervention aimed to improve wellbeing and burnout management of critical care professionals in Australia and New Zealand. The theoretical background, interventions, outcome measures and analysis plan are well detailed, except for the following:

• Page 22, lines 333-334: “However, an independent Safety Officer will be present to monitor the participant’s safety” Please explain.

• Page 25, lines 408-409: “…use of two follow-up time points in the design of this study can also establish long-term efficacy of the intervention” This sentence is questionable for a 3-month follow-up.

Reviewer #2: The present study protocol aims to assess the feasibility and user-perceived acceptability of a combined Individual

Management Plan (IMP) and Professional Problem-Solving Peer (PPSP) debrief in comparison to informal peer debrief.

I found this protocol really interesting, well-described and well-written. Also, I found the choice of the design very appropriate for the study objective.

Please find below two very minor issues which should be solved/clarified:

- please delete 'psychometric' when mentioning questionnaire/scales.

- what do you mean by 'journaling'? and by 'journal entries'?

7. PLOS authors have the option to publish the peer review history of their article (what does this mean?). If published, this will include your full peer review and any attached files.

Reviewer #1: **Yes: **ALESSANDRA SOLARI

Reviewer #2: No

---

## [Author Response · Author response to Decision Letter 0]

3 Apr 2023

Dear Professor Emily Chenette,

Re: Manuscript Re-Submission Response to Reviewers

Thank you for inviting us to resubmit our paper Manuscript ID PONE-D-22-32113, entitled Is this the solution to wellbeing and burnout management for the critical care workforce? A parallel, interventional, feasibility and realist informed pilot Randomized Control trial protocol which we would like you to consider for publication in PLOS ONE. This protocol paper summarises and contributes new knowledge and understanding to the research area of burnout and wellbeing for critical care healthcare professionals, in anticipation of an important feasibility trial 

My research team and I appreciate your advice and have made revisions according to the reviewers' comments. We have provided a response to the reviewer’s comments in the ‘Response to Reviewer’ document and have resubmitted a revised version of the manuscript (marked and un-marked). This paper has been formatted according to the PLOS ONE Guidelines.

We feel that the manuscript has significantly improved based on the reviewer’s comments and hope that the reviewers will be satisfied with our responses. Further, we confirm that:

• This manuscript is original work, and no part of the manuscript has been published or submitted elsewhere for publication. 

• All authors have contributed to developing the ideas, writing, and/or final manuscript review. 

• All authors have read and approved this version of the manuscript and its submission to the journal. 

We thank you for your consideration and look forward to hearing from you.

Yours sincerely, 

Nurul Bahirah Binte Adnan RN, BSN (Hons), PhD Candidate (On behalf of the co-authors)

College of Nursing and Health Sciences

---

## [Decision Letter · Decision Letter 1]

13 Apr 2023

Is this the solution to wellbeing and burnout management for the critical care workforce? A parallel, interventional, feasibility and realist informed pilot Randomized Control Trial protocol

PONE-D-22-32113R1

Dear Dr. Adnan,

We’re pleased to inform you that your manuscript has been judged scientifically suitable for publication and will be formally accepted for publication once it meets all outstanding technical requirements.

Kind regards,

Ali B. Mahmoud, Ph.D.

Academic Editor

PLOS ONE

Additional Editor Comments (optional):

Reviewers' comments:

Reviewer's Responses to Questions

**Comments to the Author**

1. Does the manuscript provide a valid rationale for the proposed study, with clearly identified and justified research questions?

Reviewer #1: Yes

Reviewer #2: Yes

2. Is the protocol technically sound and planned in a manner that will lead to a meaningful outcome and allow testing the stated hypotheses?

Reviewer #1: Yes

Reviewer #2: Yes

3. Is the methodology feasible and described in sufficient detail to allow the work to be replicable?

Reviewer #1: Yes

Reviewer #2: Yes

4. Have the authors described where all data underlying the findings will be made available when the study is complete?

Reviewer #1: Yes

Reviewer #2: Yes

5. Is the manuscript presented in an intelligible fashion and written in standard English?

Reviewer #1: Yes

Reviewer #2: Yes

6. Review Comments to the Author

You may also provide optional suggestions and comments to authors that they might find helpful in planning their study.

Reviewer #1: I am fine with the revised version of the manuscript. I have no further comments and congratulate with the authors.

Reviewer #2: I've read the revised version of the manuscript and I do confirm that author have provided satisfactory responses to my minor points

7. PLOS authors have the option to publish the peer review history of their article (what does this mean?). If published, this will include your full peer review and any attached files.

Reviewer #1: **Yes: **ALESSANDRA SOLARI

Reviewer #2: **Yes: **Andrea Giordano

---

## [Editor Report · Acceptance letter]

19 Apr 2023

PONE-D-22-32113R1 

Is this the solution to wellbeing and burnout management for the critical care workforce? A parallel, interventional, feasibility and realist informed pilot Randomized Control Trial protocol 

Dear Dr. Adnan:

I'm pleased to inform you that your manuscript has been deemed suitable for publication in PLOS ONE. Congratulations! Your manuscript is now with our production department. 

Kind regards, 

on behalf of

Dr. Ali B. Mahmoud 

Academic Editor

PLOS ONE